# The Future of Online Barrier-Free Open Space Cultural Experiences for People with Disabilities in the Post-COVID-19 Era

Jin-Wook Lee 

School of Plant Science & Landscape Architecture, Landscape Architecture, Hankyong National University, Anseong 17579, Gyeonggi-do, Republic of Korea; ljw@hknu.ac.kr; Tel.: +82-31-670-5212

**Abstract:** This study examines the current state of barrier-free online content in Korea and proposes strategies to revitalize online cultural experiences for individuals with disabilities. By scrutinizing existing content and conducting interviews with relevant stakeholders, the study identified prevailing challenges and potential avenues for improvement. This research suggests the following directions. First, content creation should involve soliciting input from individuals with disabilities, with an emphasis on generating experiences that reflect the daily lives of those without disabilities. Additionally, the development of diverse and convergent content, such as for educational and therapeutic functions, is crucial to cater to various user groups. The study underscores the importance of formatting content in consideration of the physical characteristics of individuals with disabilities. For sustained and efficient utilization, content must be created in a universally accessible format, accommodating users with and without disabilities. It is recommended to set various options within a single piece of content, fostering inclusivity across various disability types. Regarding content creation technology, it is crucial to utilize various methods, such as VR (virtual reality), drone filming, and virtual simulation.

**Keywords:** virtual reality; sustainable content; barrier-free open space; cultural rights of persons with disabilities



## 1. Introduction

*Study Background and Objectives*

The Internet has transformed the way people communicate [1,2], affecting their psychological well-being, individual community formation, and social identity [3]. These developments and their impacts create opportunities and barriers to social inclusion and equality among people with disabilities. In 2016, the United Nations adopted guidelines to regulate web accessibility across the public sector to make the Internet more accessible to people with disabilities. However, despite these efforts, a digital divide [4] exists that varies according to disability status, age, gender, education, and financial status [5].

Meanwhile, the proliferation of non-face-to-face environments during the pandemic has lowered barriers to cultural experiences for people with disabilities. While the cultural community was concerned that the coronavirus pandemic would lead to the demise of performing arts genres characterized by physicality, the disability community welcomed the expansion of cultural opportunities for people with disabilities. The coronavirus pandemic provided an opportunity for all genres of performing arts to move to online theaters and maximize non-face-to-face experiences using virtual spaces. This has opened opportunities for people with disabilities to experience culture and the arts indoors without having to go outside, and barrier-free methods for people with disabilities have begun to be introduced into non-face-to-face environments. The coronavirus pandemic has brought the public nature of culture and the arts to the forefront by expanding opportunities for people with disabilities to experience culture online. We observe a digital gap for people with

disabilities owing to demographic and social factors, but the content and format of Internet content are currently trying to reduce this gap. This study began with the same problem.

While barrier-free accessibility has traditionally been limited to film production and screenings, it has recently spread to theaters, exhibitions, travel, and so on. It is now possible to experience parks online. For example, the Seoul Culture Portal and National Park Service offer virtual reality (VR) services. It is now possible to have the same level of cultural experience at home as in offline spaces, and as a result, opportunities for cultural enjoyment for people with disabilities who have difficulty finding accessible open spaces have expanded. Therefore, it is now necessary to discuss how to enjoy culture through barrier-free online services in consideration of consumers who have been marginalized by cultural experiences, such as those with disabilities and impaired mobility.

The online experience space, which was a workaround prepared by creators during the coronavirus pandemic, provided direction regarding the right to enjoy culture without boundaries. If people with disabilities find it difficult to visit cultural experience spaces in person, they can now be supported online. After COVID-19, non-face-to-face cultural experiences have continued with the development of technology; however, the cultural enjoyment rights of people with disabilities may be forgotten. We must continue to pay attention to and set directions for the cultural rights of people with disabilities.

Notably, in this process, we should not emphasize the urgency of expanding barrier-free systems, but rather seek direction from the voices of those who have created, developed, and enjoyed barrier-free content (creators, developers, barrier-free service providers, users, etc.). In addition, we should objectively examine the pros and cons of online barrier-free spaces and find alternative ways to realize public cultural enjoyment in the post-COVID-19 era based on the opinions of various stakeholders.

In this regard, the purpose of this study is to identify the current status of online barrier-free content in Korea and explore ways to revitalize it. To this end, I will identify the current state of open space cultural experience content; collect opinions from various stakeholders, such as users, content creators, and people with disabilities; and explore ways to revitalize contents through various cases.

## 2. Theoretical and Literature Review

### 2.1. The Concept of Barrier-Free

In the dictionary, barrier-free refers to the movement and removal of physical and institutional barriers to make society more habitable for people with disabilities and older adults. Since the 1974 report on barrier-free design was released by the United Nations World Conference of Experts on the Living Environment of Persons with Disabilities, the concept of barrier-free design, which was used in architecture to remove barriers to social life for people with disabilities, has been expanded to include not only the physical environment but also institutions, communication, culture, and other fields [6]. Therefore, the National Theater of Korea, which produces barrier-free performances, defines barrier-free performances as "performances that are enjoyed by everyone regardless of disability".

The online barrier-free concept used in this study refers to online cases that start with the purpose of the right to cultural enjoyment of disabled audiences and deals with open spaces that are not labeled as barrier-free but have the potential to be enjoyed across the boundaries of disability and non-disability. To summarize, the scope of this study includes both online content that is barrier-free from the outset and online content that has the potential to be barrier-free. Regarding the latter, the concept of "barrier-free online open spaces" is an online extension of the traditional concept of barrier-free, which refers to accessibility in spaces outside of cities without physical barriers. In other words, barrier-free online open spaces mean that people with and without disabilities can enjoy cultural experiences in open spaces, regardless of the type of disability.

## 2.2. Review of the Relevant Legislation

Laws that may be relevant to this study include the Act on Ensuring the Convenience of the Disabled, the Elderly, and Pregnant Women (abbreviated as the Act on the Convenience of the Disabled); the Act on Promoting the Mobility of the Transportation Disadvantaged (abbreviated as the Act on the Transportation Disadvantaged); and the Act on Supporting Cultural Activities of Artists with Disabilities (abbreviated as the Act on Supporting Artists with Disabilities); as well as the Basic Act on Culture, which may be the premise for these laws.

According to Article 4 of the Basic Act on Culture, all citizens have the right to enjoy culture without discrimination in cultural expression activities. In addition, the revised Basic Act on Culture, which came into effect on 11 September 2021, includes "Matters concerning the right of access to cultural activities for persons with disabilities" in Article 8, Paragraph 3, Article 6, No. 2. This laid the foundation for strengthening accessibility for people with disabilities through cultural policy formulation and business. Various policy proposals are required to eliminate the discrimination caused by disabilities in cultural and artistic activities.

This Act ensures that people with disabilities can safely and conveniently use and access facilities. Therefore, online cultural experiences can help people with disabilities access cultural spaces offline more safely by providing information on outdoor and indoor cultural spaces in advance. To this end, it is necessary to include "building an online external space environment" in the amenities and target facilities. In addition, efforts should be made to develop non-face-to-face content and education based on Article 14 to promote research and development and to conduct education.

The Traffic Vulnerable Persons Act seeks to ensure the rights of vulnerable persons, such as the disabled and the elderly, to move safely and conveniently in all modes of transportation and on roads. The transportation facilities designated by the Act include transportation, passenger facilities, and roads. Therefore, I propose including the mobility facilities specified in the Act in online environments so that people with disabilities can move conveniently even when using roads online.

Finally, the Act on Support for Artists with Disabilities, which supports artists with disabilities, does not directly affect the construction of online barrier-free environments but can provide opportunities for artists with disabilities to contribute to the construction of online environments. If artists with disabilities participate in the development of content, they can fill in gaps that may be missed by people without disabilities. If online cultural experience content is viewed from the perspective of fostering people with disabilities as cultural artists, it can be supported through this legislation.

## 2.3. Literature Review

The discussion of barrier-free and disabled access to the arts can be categorized as follows: First, some studies have proposed measures to increase the accessibility of cultural activities to people with disabilities. In this regard, I find cases that analyze the current status of cultural leisure activities for people with disabilities, examine the current status and problems of cultural leisure activities for people with disabilities based on the viewing experiences of people with disabilities in the targeted places, and propose alternatives [7]. Additionally, I include discussions that highlight the actual situation of enjoying culture and the arts, such as performing arts and park experiences, or discussing viewing desires and proposing directions [8–14]. Additionally, discussions exist on increasing cultural accessibility by analyzing the socio-relational causes of Internet accessibility among people with disabilities [5].

The second is a review of the laws and regulations related to barrier-free services for people with disabilities. Specifically, a study was published that reviewed the laws and regulations for creating a barrier-free environment for people with disabilities and examined the positions of experts in various fields related to this issue [15]. In addition, some studies have reviewed national laws, pointed out issues that should be considered in

domestic laws, and discussed the overall direction of cultural policy support for people with disabilities [16].

Third, some studies related barrier-free access to recent changes in the media environment. Specifically, one discussion emphasizes the need for a disability perspective in the study of video culture concerning changes in the media environment centered on OTT services [15]. There has also been research on the accessibility and utilization of digital information by people with disabilities. There have been studies that have examined the current state of digital access for people with disabilities by discussing the level of the digital divide [17,18]. Subsequently, there have been policy discussions to improve this limited access [19–21]. There have also been studies on the preferences of people with disabilities in using electronic information [22] and studies analyzing the impact of digital literacy on the social isolation of people with disabilities to provide empirical evidence for policy [23]. More recently, the utilization of the metaverse has been discussed through the characteristics of how people with disabilities experience the metaverse [24].

Fourth, this study draws policy implications for performing arts-support projects that utilize virtual-space technology. Specifically, the practical challenges that performing artists face in utilizing virtual-space technologies are discussed [25].

Fifth, barrier-free spaces were applied to physical spaces. Barrier-free urban areas, landscapes, and architecture have created a barrier-free environment while using facilities and spaces. Related studies have been conducted to create barrier-free physical spaces [26–32].

Finally, there are studies on the self-determination of cultural enjoyment by people with disabilities. There are studies on the factors that limit the right to self-determination [33,34] and studies on the ability of people with disabilities to enjoy culture and ways to improve their self-determination [35–39]. In order to improve the quality of life of people with disabilities, it is effective and important to provide opportunities for self-determination [34], and it is suggested to organize programs that take into account the characteristics of disabilities and improve self-determination using repeated training and various methods [35]. It has been proven that participation in cultural activities can improve the quality of life and satisfaction of people with disabilities, enabling them to enjoy a more humanized life [40–42] and position them as members of the community through improved confidence and independence.

Regarding the right to the enjoyment of culture and the arts by persons with disabilities, measures to improve access to culture and the arts for persons with disabilities, a review of laws and regulations, the utilization and improvement of space, support projects, changes in the media environment, and recent discussions on barrier-free planning have been presented.

In the context of this research topic, barrier-free research in the external environment is discussed to create barrier-free environments in physical spaces. However, no research has been conducted on online cultural experiences or barrier-free production. This is because a barrier-free culture related to performing arts and open-space experiences has not spread sufficiently. Additionally, limited attention has been given to barrier-free online productions, which increased during the coronavirus crisis. A study drew policy implications for performing arts-support projects that utilize virtual-space technology, although it was not a barrier-free study [25]. However, while this study explores the policy implications of support projects for the performing arts using virtual-space technology through the perceptions of performing artists who use virtual-space technology, it does not focus on disabled people. This study focuses on the utilization of virtual spaces by people with disabilities and suggests alternative solutions for the right to cultural enjoyment by considering the limitations and improvement measures felt by stakeholders.

## 3. Materials and Methods

### 3.1. Study Subject

In urban spaces, the concept of barrier-free space is used to create an environment where vulnerable people, such as those with disabilities, the elderly, and children, can

use and access physical spaces without difficulty. The concept of barrier-free space in the era of COVID-19 suggests the possibility of expanding the concept of physical-space accessibility to online spaces. This could be one way to mitigate the problems of inequality and imbalance in the use of space that will arise in the post-COVID-19 era.

Therefore, this study surveyed cultural experiences utilizing virtual spaces in the urban, landscape, and architectural fields. Online barrier-free experiences have begun in more than just spatial areas (urban, landscape, and architectural spaces). Virtual-space experiences are utilized by installing booths inside buildings to introduce spaces. Recently, however, the Seoul Metropolitan Government provided a service to experience Yongsan Park, which will be built in the future, in a VR format, and the Seoul Culture Portal provides VR for spaces in Seoul such as Namsan. Under the Ministry of Environment, the National Park Service provides 54 videos of 15 parks through the 'National Park Virtual Reality (VR) Service' for people with disabilities who have transportation challenges. Given these trends, I closely examined the support and usage of cultural welfare-related organizations, content developers, and users in Seoul, and examined the experiences and needs of those involved through interviews.

Next, I examined the legal and institutional support systems for people with disabilities to experience open-space cultural activities online. Currently, Korea has implemented a barrier-free certification system to create an environment where people with disabilities, the elderly, and pregnant women can access and use cities, parks, and transportation without inconvenience, according to the Act on the Promotion of Convenience for the Transportation Disadvantaged and the Act on Ensuring the Promotion of Convenience for the Disabled, Elderly, and Pregnant Women. However, its scope is limited to physical access within spaces. Therefore, this study examines barrier-free design guidelines from overseas and identifies how the concept of barrier-free design is expanding in environments where non-face-to-face experiences are increasing.

It also explores the technical and institutional limitations of open-space experiences using VR. Experiences with virtual reality have revealed problems such as long working hours, dizziness in the virtual space, and difficult operation. By identifying difficulties in the implementation and use of real-world virtual spaces, the research team hopes to shed light on issues that need to be addressed regarding barrier-free online access for people with disabilities. Moreover, I discuss various issues that must be addressed by people with disabilities who have experienced open spaces in virtual reality.

*3.2. Study Methods*

The purpose of this study is to identify the current status of online barrier-free content in Korea and explore ways to revitalize it. To this end, I identify current barrier-free content and explore the direction of barrier-free content through the opinions of users (people with disabilities), creators, and operators. I also explore ways to incorporate it into cultural experiences through similar studies of disability-related content. Finally, I examine disability-related laws and regulations to explore institutional methods to promote online barrier-free content production(see Table 1).

**Table 1.** Research process.

| | Research Content | Research Method | Research Material |
|---|---|---|---|
| Survey | Researching currently operational websites | Search for websites on Google using the keywords open space and VR and select websites that are still in operation | Seoul Culture Portal, Seoul Yongsan Park Gallery VR Archive, Seoul Grand Park, Seoul Grand Park Botanical Garden, Bukhansan National Park, Look 360 (Selecting six websites) |
| | | ▼ | |

**Table 1.** *Cont.*

| Research Content | Research Method | Research Material |
|---|---|---|
| Analyzing Contents | Analyze from the perspective of stakeholders | Informal and in-depth interviews | Interview material (Operator, User, Content Developer) |
| | Research similar cases | Search for websites on Google and literature review | - Overseas examples of open-space virtual spaces<br>- Healing content for the disabled (Selecting nine cases) |
| | Review legislation | Review of laws related to persons with disabilities in Korea | Act on the Convenience of Persons with Disabilities/Act on the Transportation Disadvantaged/Act on Supporting Artists with Disabilities/Basic Act on Culture |

### 3.2.1. Analyzing Contents

This study analyzes both the thematic and formal features of barrier-free content. Interviews with VR content experts were conducted to analyze the characteristics of online open-space content. The data were collected and analyzed by focusing on the content operated and managed mainly by the Seoul Metropolitan Government(see Table 2).

**Table 2.** Content list.

| Order | Lists | Image |
|---|---|---|
| 1 | Seoul Culture Portal [43] |  |
| 2 | Seoul Yongsan Park Gallery VR Archive [44] |  |
| 3 | Seoul Grand Park [45] |  |
| 4 | Seoul Grand Park Botanical Garden [46] |  |

**Table 2.** *Cont.*

| Order | Lists | Image |
|:---:|:---:|:---:|
| 5 | Bukhansan National Park [47] |  |
| 6 | Look 360 [48] |  |

3.2.2. Interviews with Stakeholders

I conducted interviews with representatives of accessibility organizations and stakeholders(see Table 3). First, barrier-free content creators and developers were interviewed. Through their experiences in developing, producing, and operating barrier-free programs; problems with barrier-free support in Korea; and their opinions on the opportunities and expansion of online cultural experiences, the paper is reflected on the current state of online barrier-free cultural experiences and explored directions for cultural enjoyment in the post-COVID-19 era.

**Table 3.** Interviewee list.

| | Interviewee | Character |
|:---:|:---:|:---:|
| 1 | | Administrator of the Seoul Culture Portal [43] |
| 2 | | Administrator of Seoul Yongsan Park Gallery VR Archive [44] |
| 3 | Operator | Administrator of Seoul Grand Park [45] |
| 4 | | Administrator of National Park [46] |
| 5 | | Administrator of LOOK 360 [48] |
| 6 | | OO Shin, Central President of the Korean Association for Disability Culture/Disabled |
| 7 | User | OO Lee, Korea Association of Welfare Institutions for Persons with Disabilities |
| 8 | | OO Jo, Hearing Impaired |
| 9 | Content Developer | OO Choi, CEO of Emotion Playground, Psychotherapy Contents [49] |
| 10 | | OO Ahn, Engineer of LOOK 360 [48] |

Regarding open-space cultural activities, I contacted operators of online open-space cultural experiences in Korea, the president of the Korea Disability Culture Association, and VR content experts. Regarding online cultural experiences in Korea, I contacted representatives from the Seoul Culture Portal, VR representatives from the Yongsan Park Gallery in Seoul and Seoul National University, and TV representatives from the National Park Service to discuss the purpose of content development, content features, and operations.

Among them, OO Shin, the Chairman of the Korea Disability Culture Association, which promotes the cultural rights of people with disabilities, conducts cultural education and cultural exchange cooperation projects for marginalized groups such as the disabled. OO Shin provided direction for this study because he collected various voices of people with disabilities in the field. OO Choi, CEO of Emotion Playground and a VR content expert, has produced psychotherapy VR content such as 'Conversation of the Wind', 'Bamboo

Nokwon', and 'Suicide Prevention Bus', as well as content for healing 5.18 trauma, and operated programs. In light of this, I looked at opinions on how to utilize VR content and its possibilities.

Data were collected through notes and recordings with the interviewees' consent, and an open-ended semi-structured form was used to assess the participants' subjective feelings and thoughts. Therefore, the interviews were conducted flexibly in the field based on the interviewees' responses to the key questions.

### 3.2.3. Researching Similar Cases

I analyzed disability-related content, explored ways to incorporate it into cultural experiences, and explored ways to provide cultural enjoyment to people with disabilities by researching the types and contents of programs for people with disabilities(see Table 4).

**Table 4.** Case list.

| Order | Lists | Image | Characteristics |
|---|---|---|---|
| 1 | American Society of Landscape Architects [50] |  | - Promote and experience recently completed parks<br>- English voice support<br>- Screen adjustments with the mouse<br>- See what real park visitors are doing |
| 2 | Grand Canyon Virtual Tour [51] |  | - Enables virtual travel with VR<br>- Screen adjustments with the mouse<br>- Provides sound effects |
| 3 | Royal Park VR [52] |  | - Created for research and education<br>- Filmed using a drone<br>- Uses virtual graphics to illustrate improvements<br>- English voice support |
| 4 | Pollinator Park [53] |  | - Educates on ecosystem conservation<br>- Raises awareness of the importance of ecosystems by building virtual spaces |
| 5 | Crytek [54] |  | - Climb the world's most famous mountains virtually<br>- View the landscape through VR glasses and climb using a remote control<br>- Features realistic graphics that make you feel like you are actually up there |
| 6 | Floreo [55] |  | - VR content for autism treatment<br>- Learning modules that take into account the characteristics of autism<br>- Training to perform daily activities based on interactive scenarios |

**Table 4.** *Cont.*

| Order | Lists | Image | Characteristics |
|-------|-------|-------|-----------------|
| 7 | Hanuri Information and Culture Center [56] |  | - Uses virtual reality to treat children with developmental disabilities and brain degeneration |
| 8 | Korea Spinal Cord Injury Association [57] |  | - Created for people to experience and learn wheelchair skills virtually |
| 9 | Emotion playground [49] |  | - Uses VR for psychotherapy programs<br>- Uses sound and storytelling in VR for psychotherapy |

### 3.2.4. Reviewing Legislation

I reviewed legislation relevant to the right to cultural enjoyment for persons with disabilities and the proliferation of barrier-free online content creation. Among the laws that may be relevant to this study are the Act on Ensuring the Convenience of Persons with Disabilities, the Elderly, and Pregnant WomeLjw@n (abbreviated as the Act on the Convenience of Persons with Disabilities); the Act on Promoting the Mobility of the Transportation Disadvantaged (abbreviated as the Act on the Transportation Disadvantaged); and the Act on Supporting Cultural Activities of Artists with Disabilities (abbreviated as the Act on Supporting Artists with Disabilities); as well as the Culture Basic Act, which may be the premise for these laws. Currently, Korea is implementing a barrier-free certification system to create an environment in which people with disabilities and the elderly can access and use cities, parks, transportation, etc., without any inconvenience. I examined the possibility of expanding the concept of being barrier-free in an environment in which non-face-to-face experiences are increasing through legislation.

## 4. Results and Discussion

### 4.1. Characteristics of Online Open-Space Content

Looking at the cultural experience content of online open spaces in Korea, many contents targeting tourist destinations exist, such as museums and zoos. In the case of the Seoul Culture Portal, tourist events such as performances, exhibitions, and festivals are the main focus, while in the case of Seoul National University Park, botanical gardens and zoos are provided as content. Some sites are used for archival purposes, such as the Yongsan Park Archive, or for promoting a specific location, such as LOOK360. Most content has the advantage of allowing users to experience a specific place or event online, but this is limited to non-routine content. However, in the case of LOOK360, it is possible to obtain a glimpse of the daily lives of various people with disabilities, including not only through culture and art, but also at restaurants, cafés, bars, and academies.

On the other hand, most of the online cultural content targeting open spaces in Seoul is operated by public organizations such as the Seoul Metropolitan Government and the National Park Service, while LOOK360, as a joint stock company, targets open spaces nationwide regardless of the specific region. Among them, the content operated by public organizations has a disadvantage in that it is limited depending on the nature of the site,

and the content and format of the site operated by a private company can vary depending on the intention of the client who wants to entrust the promotion.

Regarding the content's format, a virtual separation exists between non-disabled and disabled content. Most of the current content is inaccessible to people with disabilities. Rather than explaining how a screen works, audio materials often provide background music or describe the content on the screen. Online audio is available, but it is difficult for people with visual impairments to use it because they have to look at the screen and click to hear it. Subtitles rarely contain text descriptions, and the text is not sufficiently large for easy recognition. Hence, these tools are not for people with disabilities to view the current content in an accessible way. LOOK360 provides content called a barrier-free program, which provides information on accessible restrooms, wheelchair-accessible routes, etc., in case a person with a mobility disability visits a destination. However, this content was sponsored by the Seoul Tourism Organization, and the number of videos is limited. Additionally, instructions are not available in audio or text, which is inconvenient for people with disabilities. There are characteristics of barrier-free content in online open spaces (see Table 5).

**Table 5.** Features of barrier-free content in online open spaces.

| | Purpose | How to Deliver Content | Content | Operating Entity | Features |
|---|---|---|---|---|---|
| ① Seoul Culture Portal [43] | To collect and provide VR content from local governments under the Seoul Metropolitan Government | Text, photos, voice (depending on municipality) | Ecological parks, exhibitions, festivals, and concerts | Cultural Policy Division, Seoul Metropolitan Government | A portal that collects videos produced by local governments |
| ② Seoul Yongsan Park Gallery VR Archive [44] | To record the Yongsan Park Gallery's exhibition online and provide information to a large number of people | Text, photos | Yongsan Park Gallery | Seoul Metropolitan Government Strategic Planning Division | No VR mode, direct click-and-go navigation in roadmap format |
| ③ Seoul Grand Park [45] | To provide an online view of the Seoul National University Park Zoo | Text, photos | Zoo | Seoul Grand Park | - Supports aerial view and VR mode, with pop-ups for walking perspective<br>- 360-degree videos that can be manipulated with the mouse, but do not support VR mode<br>- Does not support videos of all animals |
| ④ Seoul Grand Park Botanical Garden [46] | To provide an online view of the Seoul National University Botanical Garden | Text, photos | Botanical Gardens | Seoul Grand Park | - You cannot move around freely; you can only go to designated areas<br>- Descriptions of the plants are provided, so you can learn more about them than if you were there<br>- The image quality is very good, giving you a good sense of the scene |
| ⑤ Bukhansan National Park [47] | Created to share content with many people | Text, photos, voice | National Parks | National Park Service Office of Public Affairs | - Produces interpretive programs for people with disabilities<br>- National Parks TV provides videos on interpretive services, hearing services, diagrams, etc.<br>- No captions, but English subtitles available |
| ⑥ LOOK360 [48] | To promote and inform people about spaces | Text, photos, voice | Restaurants, cafes, bars, parks, etc. | Indispot Inc. | - Provides VR program services for people with disabilities with various themes<br>- For barrier-free programs, voice-over explanations are added to enhance user convenience<br>- Provides information on wheelchair-accessible routes and accessible restrooms<br>- The program is more of an introduction to tourist spots than a VR tour |

According to interviews with the stakeholders of the six content programs studied in this research, none were created to provide content during the COVID-19 pandemic but, rather, to provide information to and inform a wide range of people rather than cater to specific groups, such as people with disabilities. However, in the case of LOOK360, the target audience varies depending on the client or whether the project is supported by an organization or has a special purpose. Content for people with disabilities was created only with the support of these organizations.

### *4.2. Directions for Barrier-Free Cultural Experience Content in Online Open Spaces*

#### 4.2.1. What Is in the Content?

(1)  Content that is captured daily

Currently, many online cultural experiences are targeted at tourist destinations such as museums and zoos. The Seoul Culture Portal contains tourist events, such as performances, exhibitions, and festivals, and Seoul National University Park contains tourist elements of botanical gardens and zoos. Other sites include content for archival purposes, such as the Yongsan Park Archive, or for promoting a specific location, such as LOOK360. Such content has the advantage of providing users with the experience of a specific place or event but is biased toward non-routine content, such as tourism. In this regard, OO Shin, a representative of the Cultural Alliance for the Disabled, argued that content should be available every day for the disabled.

> "Everyday things that modern people do. Go to a grocery store and the market. I want to feel the routine of my day in real-time. Feeling when you open your door and leave."

> "Oh yeah, the people I live with feel this."

In the case of LOOK360, various content experiences, such as those at restaurants, cafés, bars, and academies, as well as culture and art exhibits, were organized to give a glimpse into the living spaces of people without disabilities, but it seems difficult to sense everyday life through videos made to promote the space. In addition, some of the content targeted people with disabilities, making it difficult for them to fully enjoy the site. Considering these points, it is necessary to consider and develop a way to indirectly experience the daily lives of people with disabilities, not just inform them about the living spaces of people with disabilities. Content that allows people with disabilities to experience the lives of people without disabilities is required.

(2)  Develop diverse and convergent content based on user feedback

To activate barrier-free online content, it is necessary to develop diverse and attractive content and collect user opinions. Current cultural experience content is generally characterized by providing extraordinary experiences; therefore, it is necessary to make efforts to include content for people with disabilities. To do this, it is necessary to collect and reflect on the opinions of people with disabilities and allow them to participate directly in content development. Historically, the participation of people with disabilities has been perceived as justifiably limited to their decision-making and responsibility capacities [58]. However, recent studies have reported that there are no barriers to self-determination behaviors for people with severe or intellectual disabilities [59,60], and that self-determination skills can be expressed, communicated, and learned [37,61]. Such participation can improve the quality of life and satisfaction of people with disabilities, leading to a more humanized life [40–42], and can help them to become contributing members of their community through increased confidence and independence. Below are excerpts from Shin's remarks.

> "One of the reasons I don't use the content is that it's not what I want. I desperately want to see Hallasan in the winter. I am curious about what it looks like in winter, but there are no pictures I want, so I want to see the same content in spring, summer, fall, and winter."

"I think it would be good to ask people with disabilities and do a demand survey to see what they want because right now I do not know if it's what they want in terms of format or content, and I think if they are actively involved in the development process as developers, they can pick up on things that people without disabilities do not see."

In addition, VR can be used to experience climbing, exercise, etc., to provide a variety of experiences for both non-disabled and disabled people, and to promote physical fitness and rehabilitation. The German company Crytek [54] is a virtual experience program for climbing famous mountains in the world, where users can enjoy the scenery of famous mountains through VR glasses and climb using VR remotes. This is a good example of providing content to both disabled and non-disabled people. In Korea, VR is used to treat developmental disabilities; however, the target audience is limited to people with disabilities. Of course, these attempts are meaningful; however, in the future, it will be necessary to develop content that can encompass various groups and strive to combine attractive content with educational and therapeutic functions.

Drones can provide videos from different perspectives and simulations to deliver educational content. In addition, it is necessary to develop diverse and convergent content, such as creating a virtual space to educate students on the importance of plants and animals, as in the case of Pollinator Park [53].

### 4.2.2. Format of Content

(1) How can disability characteristics be considered?

Currently, much of the content in online open spaces is inaccessible to people with disabilities. In particular, auditory materials are limited to providing background music or explaining some of the contents on the screen rather than explaining the operation of the screen. Even if auditory materials are supported, it is difficult for the visually impaired to use them because they require clicking while looking at the screen to hear them. Subtitles are rarely accompanied by text explanations, and the size of the text is difficult for the visually impaired to recognize. In light of this, it is difficult to view online open-space content in a way that considers the characteristics of people with disabilities, and it is necessary to develop content organizations that consider the characteristics of people with disabilities. People with disabilities who do not have easy access to and use of information are limited in their level of digital access because they need help such as special information and communication devices or sign language interpretation, depending on the degree and type of disability [62]. In addition, people with disabilities lack access to digital information and are socially discriminated against because of this [19]. In order to increase such digital access, it is necessary to develop a content organization method that considers the characteristics of people with disabilities.

Blind people interact with their environments through senses other than sight. They can recognize the concepts of distance and direction through hearing, but they have difficulty recognizing size and shape [63]. As observed, the concepts of size, space, and shape are formed through the sense of touch. Therefore, in online content development, it is necessary to recognize this point and use an analogy to recognize the size, shape, etc., of the visually impaired. In addition, compared to people without disabilities, people with disabilities may have slower cognitive and reaction speeds, so it is necessary to control the speed of screen switching. Repetitive learning and stimulation can help activate the brains of people with disabilities, so it is possible to develop content that allows people to experience online culture while utilizing their hands. People with disabilities have limited opportunities to interact with society, which makes them anxious about community activities. They have less experience with outside activities, which makes them feel uneasy in new and uncontrolled environments. Therefore, it is necessary to develop content that allows users to experience simulated situations. Disabilities need to be categorized into visual, hearing, cognitive, and activity disabilities, and efforts must be made to reflect the detailed characteristics of each disability. Below are some relevant remarks by OO Shin.

"There needs to be a shift in attitudes so that people with disabilities don't just fit into cars that are manufactured for them, but that they are designed with them in mind and that there are different options available."

(2)     Integrated content

For efficient and sustainable use, future online barrier-free content should be developed in a format that can be used by people with and without disabilities and should be accessible to consumers across disability types. Currently, online open-space cultural experience content is virtually separate from that of people with disabilities. Content for people without disabilities focuses on cultural enjoyment, whereas content for people with disabilities focuses on rehabilitation and therapeutic content.

However, separate production and management require a larger budget and duplicate content wastes money and time. When creating integrated content, it is preferable to provide a variety of options within one piece of content to fully enjoy the common content, rather than separating it into accessible and non-accessible versions. This equalization of opportunity allows people with disabilities to develop self-determination [64]. OO Shin had to state the direction of the content.

"If you do it separately, you lose focus on the budget, and you keep duplicating the same thing. I think it would be good to create content that can be felt by many people, and even if it takes a while, I hope that each type of disabled person can enjoy it together."

(3)     Delivering content in different ways

During the research, I found that most domestic websites that advertised their content as available in VR did not provide VR. Instead, they provided 360-degree panoramic images controlled using a mouse. VR is highly immersive, has adjustable sounds and acoustics, and is interactive. According to Choi OO Choi, the CEO of Emotion Playground [49], who ran a psychological healing program using VR, VR is very effective at healing psychological wounds because sound, light, and movement can be controlled, and cinematic motion can be used freely.

However, it is difficult to create a video for healing purposes because of limitations, such as limited image quality and limited viewpoints, in capturing and expressing the actual landscape. VR has the disadvantages of high production costs, time, and equipment, but it is becoming increasingly popular; therefore, it may be possible to apply it to some content for healing purposes. Floreo, a US-based company, utilized VR content to treat autism [55]. The company created learning modules that take into account the characteristics of autism and developed them so that people with disabilities can train to perform daily tasks. Here is what Choi has to say about VR.

"Limitations exist in capturing real-world landscapes on video, such as limited image quality, difficulty in realizing cinematic motion, and interactivity; however, VR is intentionally controllable. Therefore, one can access it freely and provide various sounds poetically and cinematically along with the video. It is excellent for creating videos to heal the mind."

Conversely, content containing a panoramic view of the Grand Canyon [51] in the United States, operated by 'ODYSSEY VISUAL MEDIA', maximizes the grandeur of the Grand Canyon through sound effects. Based on this, future online open-space barrier-free content should not only deliver information but also introduce various techniques that can stimulate the mind. In addition, it is necessary to deliver content in various ways, such as by providing videos from various perspectives using drones and videos that predict future situations through simulations.

4.2.3. Changing Perceptions

(1)     The insurance perspective

Based on previous domestic and international cases, it is challenging to conclude whether cultural content in non-face-to-face environments has been created with significant consideration for people with disabilities. It will take considerable effort to create content for both disabled and non-disabled people without boundaries. OO Shin, a representative of the cultural solidarity of people with disabilities, asserted that it was crucial to change societal perceptions. He aimed to achieve a society in which people with disabilities did not receive support only for them. Instead of a world where government support is divided between people with and without disabilities, it should be a society in which it is natural to think about and support people with and without disabilities together in any business. For this to happen, he said, people without disabilities should think that support for people with disabilities is their insurance. Instead of thinking of disability support as a form of reverse discrimination, he said that they should think of it as an insurance policy for themselves, their families, and friends.

> "You can't think of it as reverse discrimination. You must think of it as an insurance policy for you, your family, and friends."

> "It's all about social awareness. You have to recognize that it is not a government-sponsored world, and think of it as something that is ours to do."

(2)  Shifts in sociocultural perspectives

In the future, cultural projects for people with disabilities should be supported by cultural and social perspectives and not with a welfare perspective. In recent years, many developed countries, including Korea, have moved away from the past paradigm of the "individual model", in which people with disabilities must overcome their disabilities, to the "social model", in which disability is manifested in society and is constructed by socio-political structures. There has been a gradual shift from viewing disability as a tragic product of the individual to viewing it as a social problem [65].

The content and format of online cultural content for people with disabilities may differ depending on whether they are viewed from welfare, cultural, or social perspectives. While the welfare perspective looks at support for people with disabilities in terms of equity with people without disabilities, the cultural and social perspectives focus on the healing and growth of people with disabilities. Beyond merely enjoying culture, enabling people with disabilities to grow as artists or art consumers is beneficial to them and society. OO Shin, CEO of the Cultural Alliance for the Disabled, talked about his past projects supporting people with disabilities and argued that we should support people with disabilities to grow independently. When people with disabilities can grow independently, their quality of life improves, and they can fulfill their taxpayer obligations as members of society and help the government. Below are excerpts from Shin's remarks.

> "In the past, we used to go to many facilities with rice noodles and things like that, and we did a lot of painting. Honestly, if you give them a bag of soy noodles, it is just food for a few months, but if you let them earn their own money and buy their food, if they grow into professionals, artists, or art consumers, they should be active and earn an income, so that they can make a living and have a better quality of life, and they can become taxpayers and help the government."

For this to happen, online cultural content should not only convey information but also serve as a platform for people with disabilities to grow. When creating cultural content, it is necessary to keep this in mind and develop it from a long-term perspective in terms of education.

In addition, people with disabilities need assistance to fully enjoy online cultural content. OO Shin states that we need to move away from the idea of using unnecessary budgets, think of it as an investment in the growth of people with disabilities, and look at the position of assistants from the perspective of creating new jobs. Along with this, it is also important that academia and interested organizations such as the Korean Society of Landscape Architects create content that focuses on the enjoyment of online culture

by people with disabilities, such as how the American Society of Landscape Architects (ASLA) [50] provides VR content in overseas countries.

## 5. Conclusions

This study examines the status of online barrier-free content in Korea and explores ways to revitalize it. It identifies the current status of barrier-free online work in open spaces, its problems, and future directions. This study proposes the following policy recommendations. As a result of the current status survey, it was discovered that the majority of online open-space content was not produced from a barrier-free perspective. The following conclusions were drawn through interviews with stakeholders and case studies. First, it is necessary to develop content that allows people with disabilities to feel the daily life of people without disabilities in terms of content, and to create content by collecting opinions from people with disabilities who are users. Simultaneously, it is necessary to develop diverse and convergent content, such as that with educational and therapeutic functions, to cover various groups. Regarding formal aspects, it is necessary to develop a content organization method that considers the characteristics of people with disabilities. It is necessary to categorize disabilities into visual, hearing, cognitive, and activity disabilities; examine the detailed characteristics of each disability; and reflect their needs in individual content. At the same time, for the efficient and continuous use of the content, it is necessary to develop content in a format that can be used by people with disabilities and people without disabilities, and it is desirable to set various options within a single piece of content so that users can use the content together, regardless of the type of disability. In terms of content creation technology, it is also necessary to utilize various methods, such as VR, drone filming, and virtual simulation.

Based on these findings, I propose the following policy recommendations. First, it is necessary to establish a legal basis to support the creation of online content from a barrier-free perspective. Based on the Basic Act on Culture, which added content on increasing accessibility to cultural activities for people with disabilities, the Act on Accessibility for Persons with Disabilities can include the establishment of an online external space environment as an amenity and target facility and support non-face-to-face content development and education based on Article 14, the Promotion of Research and Development, and the Conduct of Education. In addition, the Act on Transportation Disadvantaged can be interpreted as expanding the mobility facilities specified in the Act to online environments. On the other hand, the Act on Support for Artists with Disabilities, which supports artists with disabilities, does not directly affect the construction of online barrier-free environments but may provide opportunities for artists with disabilities to contribute to the construction of online environments.

In addition to establishing such a legal basis and exploring options, it is proposed that a barrier-free content production support project be established that is limited to online content. This study found that the lack of a single support project is the reason why online barrier-free content production has not been activated; it is costly to produce a barrier-free version, and the production process is time-consuming. Next, we must educate those involved in barrier-free content. For the creators of barrier-free versions of content and policymakers, it is necessary to establish the need and direction of barrier-free content by conducting various programs that start from the concept and understanding of disability and removing "barriers", such as with on-screen description, subtitling, and sign language interpretation.

The significance of this study is that it objectively examined the current situation regarding the establishment and spread of barrier-free culture through current laws and cases and conducted a comprehensive review of the current status of barrier-free culture through interviews with relevant people. Furthermore, it is meaningful to expand the right to cultural enjoyment, which has expanded through COVID-19, from the perspective of people with disabilities. However, the limitations of this study are that it did not cover the realization and implementation of policies, the cases were limited to Korea, and experts

from various fields, such as people with disabilities, social welfare, and programmers, were not included. In the future, I hope that complex and specific discussions will be conducted in depth with the participation of social welfare, disability, and administrative experts, and that the contents revealed in this study will be realized. I also look forward to research that develops content based on this study to demonstrate the effectiveness of cultural enjoyment for people with disabilities.

**Funding:** This research was funded by The Seoul Institute.

**Data Availability Statement:** All the data are contained within the paper. The interviews used in this study are private and cannot be shared in full; highlights are included in the paper.

**Acknowledgments:** This manuscript was prepared by modifying and supplementing a portion of the report by the Seoul Institute, "Boundaryless Cultural Enjoyment in the Post-Corona Era: The Present and Future of Online Barrier-Free Cultural Experiences in Seoul", written by Jeenee Jun and Jin-Wook Lee. I would like to thank Jeenee Jun, with whom we worked on the project.

**Conflicts of Interest:** The author declares no conflicts of interest.

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
