# Peer review of "The Future of Online Barrier-Free Open Space Cultural Experiences for People with Disabilities in the Post-COVID-19 Era"

_land, doi:10.3390/land13010033_

Round 1

Reviewer 1 Report

Comments and Suggestions for Authors

The study has a clear and relevant aim, the methods are adequate for the analysis. However, certain shortcomings should be addressed to publish the paper:

- the number of references is too low, section 2.2. could and should be expanded; 

- the methods should be presented in a much more detailed manner. (e.g. content analysis: the sources, the number of the analysed content, the methods for selecting materials). The connection between research steps should be described more directly and in detail.

- the limitations of the study should be presented in the Conclusions;

- the Discussion should place the results in the existing body of research. This requires more references in this section and the overall expansion of the literature list;

- the author should check the text carefully to improve clarity and spelling. (e.g. line 50: "It is now 50 possible to new parks online." - what is possible now?; lines 68-69: "the contrast between barrier-free online and offline" - I think something is missing from here; sometimes the author uses "we" e.g. line 125, while in other cases uses "I" - e.g. line 204);

- the author should pay more attention to international processes and examples, practices;

- the anonymity of interview partners should be kept;

- the authors should reconsider the structure and content of the Conclusions: the majority of part focuses on the proposed future policy and development directions. While these are obviously essential, the results should be summarized in more detail here.

Author Response

Dear Reviewer,
Thank you very much for your valuable comments. I have carefully revised my manuscript based on your insightful comments. You can find my responses to your comments in the attachment. Thank you.

Reviewer 2 Report

Comments and Suggestions for Authors

This is an interesting paper but I did not really understand the purpose of the research until I had read the conclusion.  The research question should be clearly articulated at the end of the introduction, or possibly the end ofr the literature survey.  The methods should be described in such a way that it is clear how they will elicit the information needed to answer this research question and the results give the data collected.  This would allow the conclusions to be shortened and made much more focused.  There should also be some indication of further work to be undertaken following on from the results of this research.  This might be guidelines for future developments, proposals to change legislation or anything else of relevence.

Comments on the Quality of English Language

There are some odd constructions:

line 49 do you mean view?

lines 105 and 107 are the acts the same? if so use same name

line 204 changes from third to first person

table 1 - illustrations too small

lines 247- content produced needs refs.

table 3 - diagrams too small

line 263 - the various acts need refs.

table 4 needs reformatting to fit on one page.

lines 371 on - quotes need separating from text by blank lines so that they stand out and indent.  Check quotation marks to make sure they enclose the quoted text or put it in italics.

line 445 - it is challenging ... ?

line 455 - single quotation mark

Author Response

(The authors gave the same response as above.)

Round 2

Reviewer 1 Report

Comments and Suggestions for Authors

Based on the improvements made by the author I recommend accepting the article